# Homogeneous Keys, Heterogeneous Values: Exploiting Local KV Cache Asymmetry for Long-Context LLMs

**Wanyun Cui**[*,+] and **Mingwei Xu**[*]

[*]Shanghai University of Finance and Economics
[+]MoE Key Laboratory of Interdisciplinary Research of Computation and Economics, Shanghai
University of Finance and Economics
[*]cui.wanyun@sufe.edu.cn, mingweixu@stu.sufe.edu.cn

## Abstract

Recent advances in Large Language Models (LLMs) have highlighted the critical
importance of extending context length, yet the quadratic complexity of atten-
tion mechanisms poses significant challenges for efficient long-context modeling.
KV cache compression has emerged as a key approach to address this challenge.
Through extensive empirical analysis, we reveal a fundamental yet previously
overlooked asymmetry in KV caches: while adjacent keys receive similar attention
weights (*local homogeneity*), adjacent values demonstrate distinct *heterogeneous*
distributions. This key-value asymmetry reveals a critical limitation in existing
compression methods that treat keys and values uniformly. To address the limita-
tion, we propose a training-free compression framework (AsymKV) that combines
homogeneity-based key merging with a mathematically proven lossless value
compression. Extensive experiments demonstrate that AsymKV consistently out-
performs existing long-context methods across various tasks and base models.
For example, on LLaMA3.1-8B, AsymKV achieves an average score of 43.95 on
LongBench, surpassing SOTA methods like $H_2O$ (38.89) by a large margin. Our
code can be found in this link.

## 1 Introduction

The ability to process long contexts is crucial for Large Language Models (LLMs) [13, 23]. However,
processing such long contexts poses significant challenges: pre-trained LLMs face both architectural
and computational constraints in handling extended contexts. In particular, as the context length
increases, the complexity of attention mechanisms increases quadratically ($O(n^2)$), while storage
overhead increases linearly ($O(n)$) [7].

Various approaches have been proposed to address this challenge, with KV cache compression
emerging as a promising direction [15]. These methods aim to compress the KV cache while
preserving essential information for maintaining model performance. A straightforward strategy
is to keep tokens with high historical importance (e.g., attention scores [16, 19, 20, 33]). This
approach leverages the observation that attention weights exhibit significant variation across different
tokens. Another line of work attempts to identify more general token importance rather than the
history information [28, 27, 5]. However, these approaches share a fundamental limitation: they
fail to capture certain tokens that are less important in the histroy but suddenly become critical for
subsequent predictions.

To address the information loss caused by directly discarding tokens, cache merging methods have
been proposed to merge multiple tokens into fewer representations rather than hard pruning, thereby
preserving more information [32, 26, 27]. These merging approaches implicitly assume that certain

39th Conference on Neural Information Processing Systems (NeurIPS 2025).

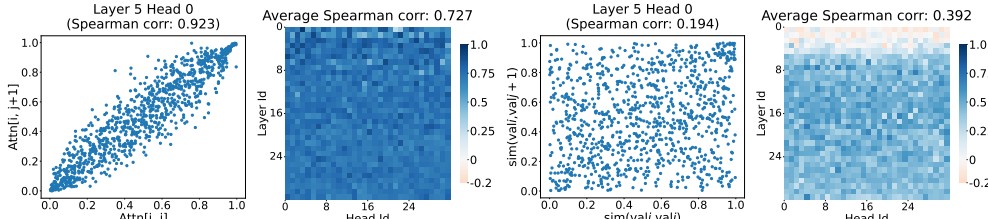

(a) Scatter plot of **key** sim-ilarity (attention) weight percentile ranks between adjacent positions

(b) Spearman correlation map of adjacent **attention** weights

(c) Scatter plot of **value** similarity percentile ranks between adjacent posi-tions

(d) Spearman correlation map of adjacent **value** similarity

Figure 1: Contrasting distributions of **local homogeneity in attentions (keys) versus local hetero-geneity in values**. Statistics are from Llama-2-7b-chat on the ShareGPT dataset. (a-b) demonstrate strong positive correlations between adjacent attention percentile ranks (normalized to [0,1], where 1 indicates highest attention) across all layers and heads, supporting the local homogeneity hypothesis for keys. (c-d) reveal weak or negative correlations between adjacent value similarity percentile ranks, computed from $\mathrm{sim}(\mathrm{val}_i, \mathrm{val}_j)$, indicating distinct heterogeneity in values. The similarity is measured by cosine. This fundamental difference between keys and values suggests the need for separate compression strategies.

redundancies or patterns exist in the KV cache. This raises a new fundamental question: *what specific characteristics of LLMs lead to these redundancies and make cache merging feasible?* We answer this question by identifying the key-value asymmetry in LLM attention mechanisms.

**Local Key-Value Asymmetry**  Through extensive empirical analysis, we reveal a fundamental pattern in attention distributions: the *homogeneity of local keys*. Specifically, we observe that **adjacent tokens consistently receive similar attention weights** - when a query assigns high attention to position $j$, the neighboring position $(j + 1)$ typically receives comparable attention weight (Fig. 1a). This pattern shows remarkable consistency across all layers and attention heads, with an average Spearman correlation coefficient of 0.727 (Fig. 1b). This consistent local attention pattern, arising from query-key interactions, suggests an underlying *homogeneity of local keys* - adjacent keys must share certain structural properties to produce such stable attention patterns. Such key homogeneity naturally emerges from language structure, where adjacent words form coherent semantic units and contribute collectively to meaning representation.

The observed homogeneity of adjacent keys provides evidence for merging neighboring tokens, offer-ing a principled explanation to recent token merging approaches. Specifically, when multiple adjacent keys exhibit high homogeneity, computing and storing only one key for the merged representation effectively approximate the original attention output, leading to both computational and memory efficiency.

However, our analysis reveals a striking asymmetry: while keys exhibit strong local homogeneity, adjacent values demonstrate markedly distinct *heterogeneous distributions*. As shown in Fig. 1c and Fig. 1d, when switching from keys to value similarities, adjacent value vectors ($\mathbf{v}_i$ and $\mathbf{v}_{i+1}$) often show much lower or even negative correlations in some layers.

This local key-value asymmetry reveals critical limitations in existing methods: cache merging methods [32, 26**?** ] apply identical merging strategies to both keys and values, overlooking their fundamentally different distributional characteristics. More studies of this phenomenon will be discussed in § A.

**Training-free Asymmetry Modeling**  Based on the above analysis, the key challenge in cache merging lies in modeling heterogeneous values. Fortunately, through careful examine of attention mechanisms' mathematical structure, we discover an elegant solution to this value heterogeneity. We develop a mathematically proven value representation scheme that guarantees lossless attention computation after merging adjacent keys. Notably, our method remains distribution-agnostic, making it inherently robust to value heterogeneity.

Building on the key-value asymmetry and the property for values, we propose AsymKV, a novel training-free cache merging method for efficient long-context modeling. Our key insight is to shift the information loss from heterogeneous values to homogeneous keys during merging, thereby minimizing overall loss. Extensive experiments demonstrate that our method consistently outperforms existing long-context methods across various tasks and base models. On LLaMA3.1-8B, AsymKV achieves an average score of 43.95 on LongBench, surpassing $H_2O$ [33] (38.89) by a significant margin. These results demonstrate AsymKV's effectiveness in extending LLMs' context handling capabilities without additional training.

**Our Contributions:** The key contributions of this work are threefold. First, we reveal a contrasting, yet previously overlooked asymmetry of local keys and values in LLM attention mechanisms. Second, based on this asymmetric property, we propose a novel training-free compression framework that combines homogeneity-based key merging with a mathematically proven lossless value representation. Third, we demonstrate through extensive experiments that our method consistently outperforms existing long-context methods across various tasks and base models.

## 2 Related Work

**KV Cache Pruning** Recent research focuses on compressing the KV cache through selective token retention and importance-based pruning. $H_2O$ [33] introduces the concept of "Heavy Hitters" - tokens that contribute significantly to attention scores - and develops a theoretically-grounded eviction policy. Building on this idea, RoCo [22] improves the robustness of cache compression by considering both temporal attention scores and stability measures. More recent works like SnapKV [16] and Scissorhands [19] leverage the persistence of token importance across generation steps, while [9] demonstrates that $L_2$ norm-based compression can achieve competitive results with a simpler implementation. However, these compression methods face a fundamental challenge: they rely heavily on token-centric measures (e.g., attention scores or norm values) to determine which tokens to retain, potentially discarding tokens that suddenly become crucial for future predictions.

**KV Cache Merging** Another line of works have explored merging similar KV cache positions to reduce memory footprint during inference. CaM [32] proposes an adaptive merging strategy guided by attention scores, while $D_2O$ [26] introduces a two-level discriminative approach considering both layer-wise patterns and token similarities. KVMerger [? ] adaptively constructs the KV cache by analyzing the intrinsic structure of attention modules. However, a fundamental limitation of these approaches is their uniform treatment of keys and values during merging despite their distinct distributional characteristics. As discovered in the introduction, this oversight is particularly problematic given the inherent heterogeneity of value vectors, which, unlike keys, often exhibit significant variations even between adjacent positions.

**Context Segmentation and Sliding** One polular variant of KV cache compression leverages context segmentation and sliding. These approaches stem from *StreamingLLM* [28], which discovered that initial tokens and recent tokens are more important than other middle tokens in the attention. Therefore, it only keeps such tokens. *LongCache* [18] expands StreamingLLM by keep critic middle tokens. These tokens are identified via the historical attention weights. *SirLLM* [31] uses token entropy to identify and keeps critic middle tokens. These segmentation-based methods require minimal KV cache operations, making them computationally efficient. However, these compression strategies essentially involve directly discarding tokens with lower weights, which results in significant information loss when these tokens suddenly become critical for future predictions.

## 3 Proposed Method

Building on the key-value asymmetry, we propose AsymKV. Our main idea is to shift the information loss from heterogeneous values to homogeneous keys during merging, thereby minimizing the loss. We show the key intuition and framework of AsymKV in Figure 2. (Left) Previous approaches that apply identical merging strategies to both keys and values suffer from significant information loss, especially considering the heterogeneous nature of values. In contrast, we leverage the asymmetry between keys and values in the attention mechanism. Our method compresses keys with minimal information loss (§ 3.1) due to their local homogeneity nature, while preserving the distinct characteristics of heterogeneous values through cardinality-aware normalization (§ 3.2).

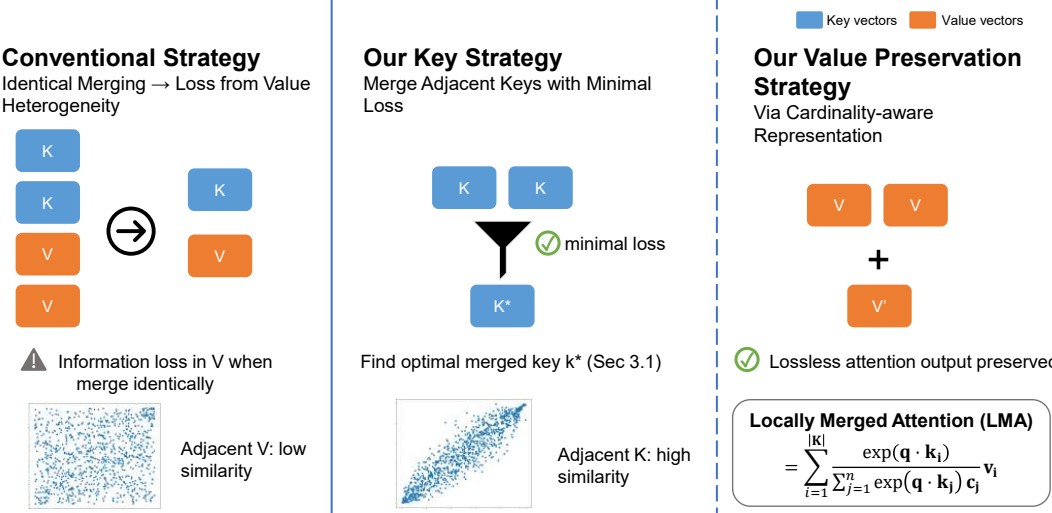

Figure 2: Illustration of our AsymKV mechanism. Left: Conventional approaches that uniformly merge both keys and values lead to information loss. Middle: We merge adjacent homogeneous keys for minimal loss. Right: We preserve their heterogeneous values through cardinality-aware normalization.

## 3.1 Homogeneous Key Merging

Our primary goal of adjacent token compression is to convert the original $n$ tokens in the KV cache into $n-1$ by merging a pair of adjacent positions $m, m+1$.

First, consider the key merging. Based on our observation of adjacent key homogeneity, we can merge adjacent keys without significantly affecting model performance. Given key vectors $\mathbf{K} = [\mathbf{k}_1, \mathbf{k}_2, \ldots, \mathbf{k}_n]$ from the KV cache and a pair of adjacent positions $m, m+1$ to be merged, let $\mathcal{L}(\mathbf{K})$ denote the language modeling loss. After compressing $\mathbf{k}_m, \mathbf{k}_{m+1}$ into one embedding $\mathbf{k}$, we denote the new loss as:

$$\mathcal{L}([\mathbf{K}_{<m}, \mathbf{k}, \mathbf{K}_{>m+1}]) \tag{1}$$

where $\mathbf{K}_{<i}$ denotes the sequence $[\mathbf{k}_1, \ldots, \mathbf{k}_{i-1}]$, and $\mathbf{K}_{>i}$ denotes $[\mathbf{k}_{i+1}, \ldots, \mathbf{k}_n]$. Our objective of optimal key compression is to find $\mathbf{k}$ that minimizes the information loss.

Due to the dimensional mismatch between the original $\mathbf{K}$ and $[\mathbf{K}_{<m}, \mathbf{k}, \mathbf{K}_{>m+1}]$ in Eq. (1) ($n \times d \rightarrow (n-1) \times d$), our cache merging takes two steps: **1.** Find a pair of identical embeddings $(\mathbf{k}^*, \mathbf{k}^*)$ to replace $(\mathbf{k}_m, \mathbf{k}_{m+1})$ while preserving dimensionality, which is mathematically tractable. **2.** Leverage attention properties to merge the two tokens.

We first find optimal embeddings $\mathbf{k}$ that minimize $\mathcal{L}([\mathbf{K}_{<m}, \mathbf{k}, \mathbf{k}, \mathbf{K}_{>m+1}])$ while keeping the dimensionality. For simplification, we denote it as $\mathcal{L}(\mathbf{k}, \mathbf{k})$, and the optimal $\mathbf{k}$ as $\mathbf{k}^*$: $\mathbf{k}^* = \arg\min_{\mathbf{k}} \mathcal{L}(\mathbf{k}, \mathbf{k})$.

We approach this optimization problem using a Newton-like method. By applying a second-order Taylor expansion of $\mathcal{L}(\mathbf{x}, \mathbf{y})$ around $(\mathbf{k}_m, \mathbf{k}_{m+1})$:

$$\mathcal{L}(\mathbf{x}, \mathbf{y}) \approx \mathcal{L}(\mathbf{k}_m, \mathbf{k}_{m+1}) + \nabla\mathcal{L}(\mathbf{k}_m, \mathbf{k}_{m+1})^\top \begin{bmatrix} \mathbf{x} - \mathbf{k}_m \\ \mathbf{y} - \mathbf{k}_{m+1} \end{bmatrix} + \frac{1}{2} \begin{bmatrix} \mathbf{x} - \mathbf{k}_m \\ \mathbf{y} - \mathbf{k}_{m+1} \end{bmatrix}^\top \mathbf{H} \begin{bmatrix} \mathbf{x} - \mathbf{k}_m \\ \mathbf{y} - \mathbf{k}_{m+1} \end{bmatrix} \tag{2}$$

where $\mathbf{H}$ is the Hessian matrix at $(\mathbf{k}_m, \mathbf{k}_{m+1})$. We denote the Hessian matrix $\mathbf{H}$ as:

$$\mathbf{H} = \begin{bmatrix} \mathbf{H}^{11} & \mathbf{H}^{12} \\ \mathbf{H}^{21} & \mathbf{H}^{22} \end{bmatrix} \tag{3}$$

Each submatrix $\mathbf{H}^{ab}$ is a $d \times d$ matrix. To minimize $\mathcal{L}(\mathbf{k}, \mathbf{k})$, we set $\mathbf{x} = \mathbf{k}$, $\mathbf{y} = \mathbf{k}$ and substitute into our quadratic approximation:

$$\mathcal{L}(\mathbf{k}, \mathbf{k}) \approx \mathcal{L}(\mathbf{k}_m, \mathbf{k}_{m+1}) + \nabla \mathcal{L}(\mathbf{k}_m, \mathbf{k}_{m+1})^\top \begin{pmatrix} \mathbf{k} - \mathbf{k}_m \\ \mathbf{k} - \mathbf{k}_{m+1} \end{pmatrix} + \frac{1}{2} \begin{pmatrix} \mathbf{k} - \mathbf{k}_m \\ \mathbf{k} - \mathbf{k}_{m+1} \end{pmatrix}^\top \mathbf{H} \begin{pmatrix} \mathbf{k} - \mathbf{k}_m \\ \mathbf{k} - \mathbf{k}_{m+1} \end{pmatrix}$$
(4)

Following the Newton method, we find the critical point by setting the gradient of this quadratic approximation to zero. This yields the optimal solution (details in Appendix C):

$$\mathbf{k}^* = (\mathbf{H}^{11} + 2\mathbf{H}^{12} + \mathbf{H}^{22})^{-1} [\mathbf{H}^{11} \mathbf{k}_m + \mathbf{H}^{12}(\mathbf{k}_m + \mathbf{k}_{m+1}) + \mathbf{H}^{22} \mathbf{k}_{m+1} - (\mathbf{g}_m + \mathbf{g}_{m+1})]$$
(5)

where $\mathbf{g}_m = \nabla_{\mathbf{k}_m} \mathcal{L}$ and $\mathbf{g}_{m+1} = \nabla_{\mathbf{k}_{m+1}} \mathcal{L}$.

To efficiently compute the Hessian matrix $\mathbf{H}$, we use the Fisher information matrix as an approximation, which is a common technique in second-order optimization methods. Following the approach in neural network pruning [12], we assume that the interactions between parameters are negligible and approximate the Fisher information matrix as a diagonal matrix. The diagonal elements can be efficiently computed using their gradients:

$$\mathbf{H}_{ii} = F_{ii} = \nabla \mathcal{L}(\mathbf{k}_m, \mathbf{k}_{m+1})_i^2$$
(6)

In our empirical analysis, we discover that the magnitude of gradient terms $\mathbf{g}_m$ and $\mathbf{g}_{m+1}$ can exceed that of $\mathbf{H}$ by six orders of magnitude in Eq. (5). This is a known issue in Newton-like methods when the function is far from its minimum or when the curvature is very small. To stabilize the optimization and maintain $\mathbf{k}^*$ as a valid key, we adopt a modified Newton approach by dropping the gradient terms from Eq. (5), effectively using only the curvature information to guide our solution.

**Compression Position Selection** To minimize information loss during merging, we select positions $m, m+1$ with the lowest sum of attention scores, where positions receiving minimal attention have the least impact on the model's attention mechanism.

### 3.2 Cardinality Normalization for Lossless Value Merging

After replacing adjacent keys with identical embeddings, two challenges remain: (1) how to reduce input tokens $(\mathbf{k}^*, \mathbf{k}^*) \rightarrow \mathbf{k}^*$ to improve computing efficiency; (2) how to merge their corresponding values. We elaborate how to extend the attention mechanism to address both challenges while maintaining output equivalence.

In the original attention mechanism, the output for query $\mathbf{q}$ is:

$$\text{Attention}(\mathbf{q}, \mathbf{K}, \mathbf{V}) = \sum_{i=1}^{|\mathbf{K}|} \frac{\exp(\mathbf{q} \cdot \mathbf{k}_i)}{\sum_{j=1}^{|\mathbf{K}|} \exp(\mathbf{q} \cdot \mathbf{k}_j)} \mathbf{v}_i$$
(7)

After key compression where $\mathbf{k}_m = \mathbf{k}_{m+1} = \mathbf{k}^*$ in § 3.1, we have:

$$\text{Attention}(\mathbf{q}, \mathbf{K}, \mathbf{V}) = \sum_{i \in [1,n] \setminus \{m, m+1\}} \frac{\exp(\mathbf{q} \cdot \mathbf{k}_i)}{\sum_{j=1}^{|\mathbf{K}|} \exp(\mathbf{q} \cdot \mathbf{k}_j)} \mathbf{v}_i + \underbrace{\frac{\exp(\mathbf{q} \cdot \mathbf{k}^*)}{\sum_{j=1}^{n} \exp(\mathbf{q} \cdot \mathbf{k}_j)} (\mathbf{v}_m + \mathbf{v}_{m+1})}_{\substack{\text{attention to one merged KV pair} \\ \text{(key=}\mathbf{k}^*\text{, value=}\mathbf{v}_m + \mathbf{v}_{m+1})}}$$
(8)

Examining Eq. (8), we observe a key insight: after converting both $\mathbf{k}_m, \mathbf{k}_{m+1}$ to $\mathbf{k}^*$, the attention output for two original tokens $m$ and $m+1$ is mathematically equivalent to the attention output for a single compressed token with key $\mathbf{k}^*$ and value $(\mathbf{v}_m + \mathbf{v}_{m+1})$.

**(Locally Merged Attention)** The insight above naturally suggests an alternated attention mechanism for merged tokens, which we denote as Locally Merged Attention (LMA):

$$\text{LMA}(\mathbf{q}, \mathbf{K}, \mathbf{V}, \mathbf{C}) = \sum_{i=1}^{|\mathbf{K}|} \frac{\exp(\mathbf{q} \cdot \mathbf{k}_i)}{\sum_{j=1}^{n} \exp(\mathbf{q} \cdot \mathbf{k}_j) \mathbf{c}_j} \mathbf{v}_i$$
(9)

where $\mathbf{c}_i$ indicates the number of original tokens represented by the $i$-th compressed token. The cardinality vector $\mathbf{c}$ is designed to maintain the denominator in Eq. (8), ensuring mathematical equivalence between the original and compressed attention mechanisms. Initially, $\mathbf{c}_i = 1$ for all tokens. After merging positions $m$ and $m+1$, we update $\mathbf{C}$ as $[\mathbf{C}_{<m}, \mathbf{c}_m + \mathbf{c}_{m+1}, \mathbf{C}_{>m+1}]$, ensuring the denominator in our attention calculation remains equivalent to the original uncompressed attention.

**(Equivalence)** Using LMA, the following equivalence holds:

$$\text{Attention}(\mathbf{q}, \mathbf{K}, \mathbf{V}) = \text{LMA}(\mathbf{q}, \mathbf{K}', \mathbf{V}', \mathbf{C}) \tag{10}$$

where $\mathbf{K}, \mathbf{V}$ are KV caches for $n$ tokens while $\mathbf{K}', \mathbf{V}'$ are for $n-1$ tokens:

$$
\begin{aligned}
\mathbf{K}' &= [\mathbf{K}_{<m}, \mathbf{k}^*, \mathbf{K}_{>m+1}] \\
\mathbf{V}' &= [\mathbf{V}_{<m}, \mathbf{v}_m + \mathbf{v}_{m+1}, \mathbf{V}_{>m+1}]
\end{aligned} \tag{11}
$$

This equivalence reveals an elegant characteristic of attention mechanisms: they permit lossless compression of values through simple vector addition. The only cost is to store $\mathbf{C}$ with $n$ integers. This property is particularly powerful as it enables preservation of attention outputs while reducing sequence length, effectively solving the quadratic complexity challenge of long-sequence processing.

### 3.3 Efficient Implementations for Long-Text Generation

#### 3.3.1 Time-Efficiency by Chunk-wise Parallel Compression

We propose a chunk-based parallel compression method for efficient long-text generation. The process predicts the next token using HLA in place of the original attention, without requiring any model fine-tuning. More specifically, when the context length reaches `max_length`, after every `chunk_size` new tokens, we compress `max_length + chunk` tokens into `max_length` tokens in parallel by: **1.** Identifying `chunk` pairs of adjacent tokens with lowest attention scores. **2.** Computing optimal compression according to Eq. (5). **3.** Merging keys, values and cardinalities.

Since the merge operation is performed only once every `chunk_size` tokens (e.g., 512 tokens), its computational overhead is minimal relative to the overall inference process. The compression step requires only a single backward pass to compute the Hessian matrices for all candidate compression positions, followed by parallel execution of the optimal compression operations. This design ensures that AsymKV maintains inference efficiency. More experimental results are shown in § 4.5.

#### 3.3.2 Memory-Efficiency by Selective Gradient Computation

Our method requires gradient computation (Eq. (5)) which might raise concerns about increased memory usage. However, AsymKV still maintains memory efficiency compared to other approaches. Unlike typical backpropagation that computes gradients for all parameters, we only compute gradients for **key embeddings within the current chunk, not for all model parameters**.

To elaborate further from a quantitative perspective, this selective gradient computation yields a gradient tensor of size approximately $c \times d$, where $c$ denotes the chunk size and $d$ is the dimension per token. In contrast, a standard forward pass (and its associated gradient computation) requires storing the full model parameters (with size $O(p)$, where $p$ is the total number of parameters) along with the KV cache states, sized at $2 \times l \times d$ (where $l$ represents the maximum sequence length). Thus, the additional memory overhead from our selective gradients is on the order of $O(cd)$, which is significantly smaller than the baseline's $O(2ld + p)$ when $c \ll l$ (a typical scenario in chunked processing). This avoids unnecessary gradient computations across the entire sequence, ensuring that both memory and computational overhead are greatly reduced. Detailed memory statistics are provided in § 4.5.

## 4 Experiments

### 4.1 Experimental Setup

**Baselines** We compare AsymKV against several categories of approaches: *KV cache compression*: H$_2$O [33], *KV cache merge*: CaM [32] ,*prompt compression*: LLMLingua-2.0 [21], *context segmentation*: StreamingLLM [28] and LongCache [18].

Table 1: Performance on LongBench. AsymKV outperforms its baselines on most settings.

| | Single-Doc | Multi-Doc | Sum | Few-shot | Synthetic | Code | Avg. |
|---|---|---|---|---|---|---|---|
| **Llama2-7B-chat** | | | | | | | |
| Full Context | 25.80 | 21.47 | 24.62 | 62.86 | 4.96 | 48.90 | 32.00 |
| StreamingLLM | 19.29 | 21.05 | 23.15 | 60.85 | 1.81 | 48.58 | 29.61 |
| LongCache | 19.73 | 20.06 | 23.19 | 61.26 | 2.24 | 49.05 | 29.71 |
| $H_2O$ | 19.92 | **25.64** | 23.85 | 61.37 | 4.27 | 50.28 | 31.34 |
| LLMLingua-2 | 21.47 | 23.29 | 23.53 | 33.23 | 6.17 | 35.12 | 24.20 |
| CaM | 19.53 | 20.64 | 22.67 | 61.81 | 4.18 | 48.53 | 30.15 |
| **AsymKV** | **24.63** | 24.15 | **24.22** | **62.11** | **10.18** | **52.16** | **33.12** |
| **Llama3.1-8B-Instruct** | | | | | | | |
| Full Context | 43.73 | 44.49 | 29.12 | 69.36 | 53.56 | 52.94 | 60.21 |
| StreamingLLM | 28.15 | 27.19 | 25.15 | 63.17 | 16.33 | 54.02 | 35.73 |
| LongCache | 28.98 | 27.84 | 25.35 | 64.73 | 19.68 | 53.60 | 36.70 |
| $H_2O$ | 33.30 | 34.43 | 26.60 | **66.23** | 14.75 | **55.56** | 38.89 |
| LLMLingua-2 | 32.02 | 32.24 | 24.99 | 27.87 | 17.67 | 52.63 | 30.75 |
| CaM | 32.14 | 32.63 | 24.91 | 63.09 | 16.77 | 54.03 | 37.49 |
| **AsymKV** | **39.42** | **38.93** | **27.30** | 65.66 | **39.39** | 55.24 | **43.95** |
| **Mistral-7B-Instruct-v0.3** | | | | | | | |
| Full Context | 38.74 | 38.29 | 29.04 | 70.70 | 51.00 | 55.06 | 46.40 |
| StreamingLLM | 24.80 | 22.14 | 25.18 | 66.49 | 15.14 | 53.51 | 34.57 |
| LongCache | 26.05 | 22.31 | 25.44 | 66.21 | 14.93 | 53.43 | 34.80 |
| $H_2O$ | 29.66 | 28.22 | 26.32 | **67.78** | 14.83 | 53.95 | 37.09 |
| LLMLingua-2 | 28.12 | 28.62 | 25.75 | 45.85 | 16.00 | 48.81 | 32.17 |
| CaM | 26.15 | 29.06 | 26.81 | 66.16 | 20.96 | 53.76 | 37.12 |
| **AsymKV** | **33.71** | **32.81** | **27.04** | 67.21 | **34.56** | 54.93 | **41.33** |
| **Qwen2-7B-Instruct** | | | | | | | |
| Full Context | 37.07 | 41.77 | 28.27 | 68.61 | 36.25 | 50.67 | 43.81 |
| StreamingLLM | 27.73 | 28.14 | 24.32 | 66.85 | 7.50 | 49.55 | 34.70 |
| LongCache | 27.98 | 28.98 | 24.80 | 66.38 | 9.00 | 48.34 | 34.94 |
| $H_2O$ | 29.91 | 28.49 | 25.21 | 68.08 | 12.25 | **53.63** | 36.68 |
| LLMLingua-2 | 30.04 | 31.71 | 24.63 | 46.32 | 6.50 | 50.09 | 31.96 |
| CaM | 29.14 | 28.59 | 25.87 | 66.33 | 9.25 | 48.88 | 35.38 |
| **AsymKV** | **33.72** | **34.46** | **26.24** | **69.88** | **14.25** | 49.33 | **38.76** |

**Base Models** To demonstrate the generality of AsymKV, we evaluate across diverse model architectures: Llama2-7B-chat [24], Llama3.1-8B-Instruct [10], Mistral-7B-Instruct-v0.3 [11], and Qwen2-7B-Instruct [29].

**Implementation Details** Unless otherwise specified, we set the compression context `max_length` to 2048 tokens and `chunk_size` to 512. All baselines use the same settings for fair comparison. For $H_2O$, we set recent budget to 2048 and heavy budget to 512. Following attention sink [28], we always preserve the initial 32 tokens. All experiments are conducted on NVIDIA A100 80GB.

## 4.2 Long Context Performance Evaluation

We evaluate AsymKV's effectiveness on LongBench [2], a comprehensive benchmark for long-context understanding. LongBench contains 16 English tasks from a wide range of categories.

**Results** As shown in Table 1, AsymKV consistently outperforms existing long-context methods across different models and tasks. On LLaMA3.1-8B, AsymKV achieves a 43.95 average score, surpassing $H_2O$ (38.89) and other baselines by a significant margin. The improvement is particularly pronounced in challenging tasks like Synthetic reasoning, where AsymKV scores 39.39 compared to $H_2O$'s 14.75. For all base models, AsymKV maintains its advantage with average scores, showing substantial improvements over baselines.

Table 2: Performance on LongBenchV2.

| Model | Overall | Easy | Hard | Short | Medium | Long |
|---|---|---|---|---|---|---|
| Full Context | 30.02 | 30.73 | 29.58 | 35.00 | 27.91 | 25.93 |
| StreamingLLM | 27.04 | 27.60 | 26.69 | 32.78 | 23.26 | 25.00 |
| LongCache | 28.43 | 28.13 | 28.62 | 32.78 | 25.58 | 26.85 |
| $H_2O$ | 28.23 | 28.12 | 28.29 | 31.67 | 26.98 | 25.00 |
| CaM | 28.23 | 28.64 | 27.97 | 31.67 | 26.98 | 25.00 |
| **AsymKV** | **30.02** | **30.23** | **29.90** | **32.78** | **27.44** | **28.85** |

Table 3: Performance on early topic retrieval.

| Model | Qwen2-7B | Llama3.1-8B | Mistral-7B |
|---|---|---|---|
| Full Context | 47.33 | 80.00 | 42.67 |
| StreamingLLM | 0.00 | 0.00 | 0.00 |
| LongCache | 0.00 | 0.00 | 8.67 |
| CaM | 12.67 | 24.67 | 15.33 |
| $H_2O$ | 21.33 | 63.33 | 38.67 |
| LLMLingua2 | 0.00 | 0.00 | 0.00 |
| **AsymKV** | **36.00** | **75.33** | **40.67** |

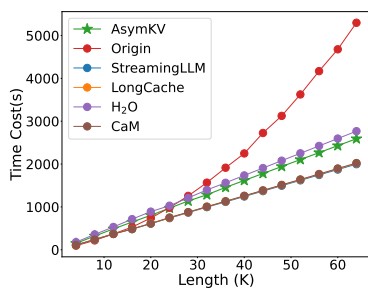

Figure 3: Inference efficiency.

**Precise Information Retrieval** In Single-Doc and Multi-Doc QA tasks, which require precise information retention, AsymKV consistently outperforms other methods by significant margins (5-10 points). This suggests that our homogeneity-based compression effectively preserves key information needed for accurate question answering.

**Extreme Long-Context Compression** We evaluate AsymKV on LongBenchV2 [3], a benchmark with contexts ranging from 8,000 to 2 million tokens across six task categories (multi-document QA, code comprehension, temporal reasoning, mathematical derivation, cross-lingual understanding, and hierarchical information synthesis). Using Llama3.1-8B-Instruct with cache_size=8192, Table 2 shows that AsymKV matches full-context methods in short contexts while significantly outperforming baselines in medium to long contexts (up to millions of tokens), demonstrating its effectiveness in extreme long-context senarios.

**Regularization Effect of AsymKV.** AsymKV often outperforms the full-context model across various tasks, suggesting that it acts as a form of regularization in long-context settings. Due to the inherent limitations of LLMs in handling extended contexts, full KV caches tend to accumulate redundant tokens with low attention scores, diluting focus on relevant information. By selectively merging these low-attention tokens, AsymKV effectively suppresses contextual noise, leading to more focused and efficient inference. A similar regularization phenomenon has also been observed in related KV-cache optimization methods [33].

### 4.3 Comprehensive Information Retention

KV cache compression methods face multiple challenges in information retention: they must preserve not only early context details but also maintain the ability to capture document-level semantic structure and sequential relationships. To evaluate models' comprehensive information retention capabilities, we conduct experiments using TopicRet [14] from L-Eval [1]. This benchmark is particularly challenging as it requires models to answer questions about *the second or third topic in multi-topic documents*, testing their ability to retain early context information.

The results in Table 3 reveal several key findings. First, context segmentation methods (StreamingLLM, LongCache) and prompt compression approaches (LLMLingua2) completely fail at this task, scoring zero or near-zero across all models. This dramatic performance drop confirms our hypothesis that discarding or imprecisely compressing early tokens severely impairs models' ability to access historical information. Although methods like CaM and $H_2O$ show some capability in retaining early information, their performance significantly lags behind full-context processing. In contrast, AsymKV demonstrates remarkable effectiveness in preserving early context information, achieving scores close to full-context processing (75.33 vs 80.00 on LLaMA3.1-8B).

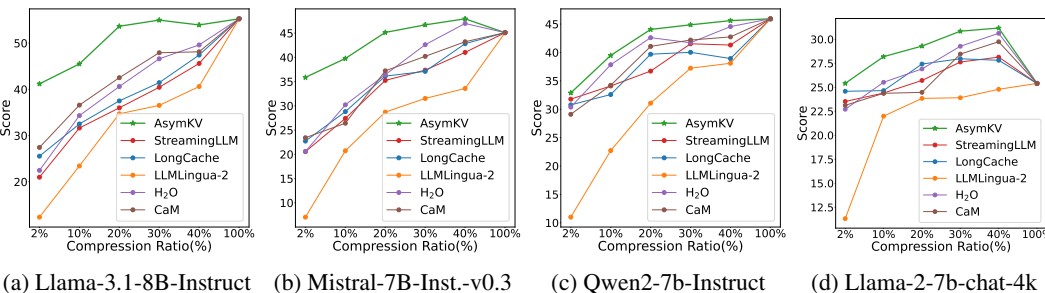

| (a) Llama-3.1-8B-Instruct | (b) Mistral-7B-Inst.-v0.3 | (c) Qwen2-7b-Instruct | (d) Llama-2-7b-chat-4k |

Figure 4: Effect of different compression ratios.

Table 4: Peak GPU Memory (MB)

| Method | L.2-7B | L.2-13B |
|---|---|---|
| StreamingLLM | 19,592 | 39,911 |
| LongCache | 24,968 | 49,236 |
| $H_2O$ | 22,479 | 47,310 |
| CaM | 22,548 | 47,476 |
| AsymKV | 24,923 | 48,671 |

Figure 5: Ablation study on different merge strategies.

| | Value Merge Strategy | |
|---|---|---|
| Key Merge Strategy | Same as Key | Asymmetric |
| Mean Merge | 18.93 | 21.86 |
| Weighted by Cardinality | 20.90 | 23.91 |
| Weighted by Attention | 14.99 | 23.83 |
| **Optimal $k^*$ (Eq. (5))** | 21.21 | **24.43** |

## 4.4 Compression Rate Analysis

To systematically evaluate AsymKV's context compression capabilities, we analyze its performance across different compression rates on the long-context HotpotQA [30] task from LongBench. Here, the compression rate is defined as the ratio between the compressed and original token counts.

As shown in Fig. 4, AsymKV demonstrates superior compression capabilities across all compression rates. Most notably, with only 20% of the original context length, AsymKV achieves performance comparable to full-context processing, significantly outperforming all baseline methods. This robust performance highlights AsymKV's effectiveness in preserving crucial contextual information even under aggressive compression.

## 4.5 Inference Efficiency

We evaluated the computational efficiency of different approaches during text generation. To do this, we had the models generate text using greedy sampling on Mistral-7B-Instruct-v0.3 and measured the time required to generate different numbers of tokens.

**Inference Speed** Fig. 3 reveals that among the evaluated methods, context segmentation approaches (StreamingLLM and LongCache) achieve the highest computational efficiency due to their minimal KV cache operations. However, this efficiency comes at the cost of performance. AsymKV strikes a better balance, achieving the highest efficiency among compression-based methods.

**Memory Consumption** We measure peak GPU memory usage on LLaMA3.1-8B-Instruct with a cache size of 2048 and chunk size of 128. As shown in Table 4, AsymKV's memory consumption is comparable to other baselines. This validates that both generation and compression costs are practical and scalable.

## 4.6 Ablations on Merge Strategies

We conduct an ablation study to compare different strategies for merging keys and values during compression. For key merging, we compare four approaches: simple mean pooling, cardinality-weighted averaging, attention score-weighted averaging, and our optimal strategy derived from Eq. (5). For each key merge strategy, we experiment with two value merge strategies: either using the identical strategy as keys, or using our proposed asymmetric cardinality-normalized method.

Results in Table 5 are scores on the multi-hop QA task MuSiQue [25] from LongBench. The results demonstrate two key findings. First, our theoretically-derived optimal key merge strategy consistently

outperforms other approaches. This empirically validates the foundation of our optimal merging strategy presented in § 3.1. Second, the results show that using distinct strategies for keys and values is beneficial.

# 5 Conclusion

In this paper, through extensive empirical analysis, we reveal a fundamental yet previously overlooked pattern: local KV cache Asymmetry. This property motivates our key technical innovation—a training-free merging framework that combines homogeneity-based key merging with mathematically proven lossless value representation. We present AsymKV, a novel approach to address the computational challenges of long-context modeling in LLMs. Our comprehensive experiments demonstrate that AsymKV outperforms existing long-context methods across various tasks and base models.

**Limitations.** Further applications of AsymKV need to consider compatibility with methods like FlashAttention and vLLM. We view this as an engineering problem and are actively working to address it.

**Acknowledgments and Disclosure of Funding** This paper was supported by the Shanghai Natural Science Foundation (25ZR1402137).

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

# A Analysis: Key-Value Asymmetry in Attentions

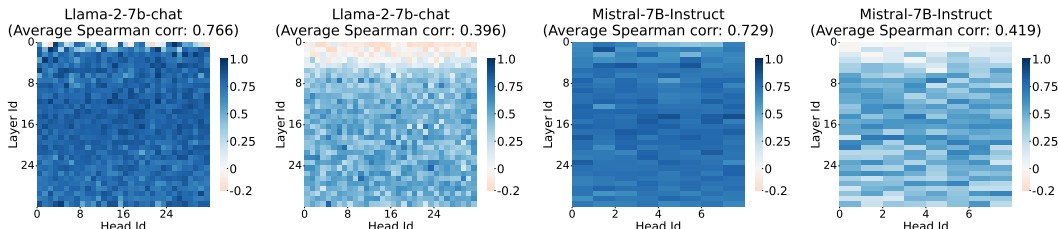

(a) Key similarity correlation heatmap on QASPER
(b) Value similarity correlation heatmap on QASPER
(c) Key similarity correlation heatmap
(d) Value similarity correlation heatmap

Figure 6: Contrasting distributions of local key distributions versus local value distributions across different datasets and model architectures. Heatmaps show Spearman correlation coefficients between adjacent tokens across layers (y-axis) and attention heads (x-axis). The consistent strong positive correlations for local keys (a,c) and weak/negative correlations for local values (b,d) suggesting these are universal properties of LLM KV caches.

Initial experiments with Llama2-7b-chat on ShareGPT [6] data (Fig. 1) revealed a striking asymmetry between local key and value distributions. To obtain a more direct observation, we analyze the Spearman correlation coefficient of $\mathrm{sim}(\mathrm{key}_i, \mathrm{key}_j)$ and $\mathrm{sim}(\mathrm{val}_i, \mathrm{val}_j)$ for different $i, j$. To validate the universality of these patterns, we conducted a comprehensive analysis across diverse settings: (1) different data distributions, including academic papers (QASPER [8]) and multi-domain questions (MultiFieldQA(en) [2]) and (2) various model architectures, specifically Mistral-7B-Instruct-v0.3 and Qwen2-7B-Instruct [29]). The results for QASPER and Mistral-7B-Instruct-v0.3 are shown in Fig. 6, with additional results for MultiFieldQA and Qwen2-7B-Instruct presented in Appendix B.

**Key Homogeneity**: Adjacent keys exhibit consistently strong positive correlations across all layers and attention heads (average correlation coefficient $> 0.7$), indicating robust encoding of local semantic relationships in key representations.

**Value Heterogeneity**: In stark contrast, adjacent values show significantly lower (average correlation coefficient $< 0.4$) or even negative correlations, suggesting that value vectors encode distinct and complementary aspects of token information. This heterogeneity appears essential for maintaining the model's representational capacity.

# B Distribution of Local Keys and Values in More Models and Datasets

To validate that our observations about the asymmetric properties of keys and values are general across different models and datasets, we conduct additional experiments on the Multifieldqa(en) dataset and the Qwen2-7B-Instruct model. The results are shown in Figure 7.

As shown in Figure 7, the key similarity heatmaps (Figure 7a, 7c) consistently exhibit strong diagonal block patterns, indicating high local homogeneity. In contrast, the value similarity heatmaps (Figure 7b, 7d) show heterogeneous distributions. These results confirm that the asymmetric properties we observed are inherent characteristics of transformer attention mechanisms rather than artifacts of specific models or datasets.

# C Solving for the Optimal Key Vector

Continue from (4):

**Gradient Term Expansion:**

$$\nabla\mathcal{L}(\mathbf{k}_m, \mathbf{k}_{m+1})^\top \begin{bmatrix} \mathbf{k} - \mathbf{k}_m \\ \mathbf{k} - \mathbf{k}_{m+1} \end{bmatrix} = \mathbf{g}_m^\top(\mathbf{k} - \mathbf{k}_m) + \mathbf{g}_{m+1}^\top(\mathbf{k} - \mathbf{k}_{m+1}) \tag{12}$$

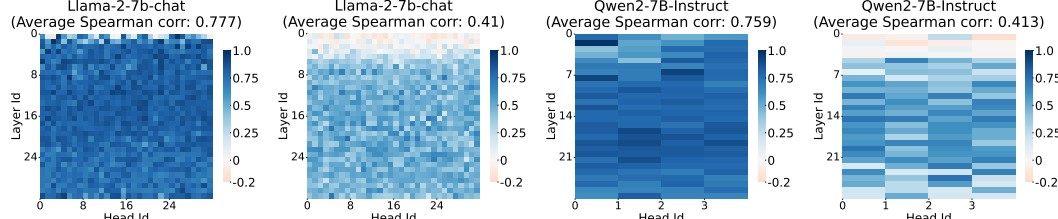

(a) Key correlation on Mul- (b) Value correlation on (c) Key correlation on (d) Value correlation on
tiFieldqa(en)        MultiFieldqa(en)       Qwen2-7B            Qwen2-7B

Figure 7: Similarity heatmaps of local keys and values across different models and datasets (LLaMA2-7B-chat on MultiFieldqa(en) and Qwen2-7B-Instruct on ShareGPT). The diagonal blocks in key heatmaps (a, c) indicate strong local homogeneity, while the more scattered patterns in value heatmaps (b, d) demonstrate heterogeneity.

**Quadratic Term Expansion:**

$$\frac{1}{2}\begin{bmatrix}\mathbf{k}-\mathbf{k}_m\\\mathbf{k}-\mathbf{k}_{m+1}\end{bmatrix}^\top \mathbf{H}\begin{bmatrix}\mathbf{k}-\mathbf{k}_m\\\mathbf{k}-\mathbf{k}_{m+1}\end{bmatrix} = \frac{1}{2}(\mathbf{k}-\mathbf{k}_m)^\top\mathbf{H}^{11}(\mathbf{k}-\mathbf{k}_m) + (\mathbf{k}-\mathbf{k}_m)^\top\mathbf{H}^{12}(\mathbf{k}-\mathbf{k}_{m+1})$$
$$+ \frac{1}{2}(\mathbf{k}-\mathbf{k}_{m+1})^\top\mathbf{H}^{22}(\mathbf{k}-\mathbf{k}_{m+1}) \tag{13}$$

**Constructing the Total Objective Function**

Adding the above terms, the objective function with respect to $\mathbf{k}$ is:

$$\mathcal{L}(\mathbf{k}) \approx \mathcal{L}(\mathbf{k}_m,\mathbf{k}_{m+1}) + \mathbf{g}_m^\top(\mathbf{k}-\mathbf{k}_m) + \mathbf{g}_{m+1}^\top(\mathbf{k}-\mathbf{k}_{m+1}) + \frac{1}{2}(\mathbf{k}-\mathbf{k}_m)^\top\mathbf{H}^{11}(\mathbf{k}-\mathbf{k}_m)$$
$$+ (\mathbf{k}-\mathbf{k}_m)^\top\mathbf{H}^{12}(\mathbf{k}-\mathbf{k}_{m+1}) + \frac{1}{2}(\mathbf{k}-\mathbf{k}_{m+1})^\top\mathbf{H}^{22}(\mathbf{k}-\mathbf{k}_{m+1}) \tag{14}$$

**Taking the Derivative of $\mathcal{L}(\mathbf{k})$ with Respect to $\mathbf{k}$ and Setting to Zero**

Taking the derivative:

$$\frac{\partial\mathcal{L}}{\partial\mathbf{k}} = \mathbf{g}_m + \mathbf{g}_{m+1} + \mathbf{H}^{11}(\mathbf{k}-\mathbf{k}_m) + \mathbf{H}^{12}(\mathbf{k}-\mathbf{k}_{m+1}) + \mathbf{H}^{12\top}(\mathbf{k}-\mathbf{k}_m) + \mathbf{H}^{22}(\mathbf{k}-\mathbf{k}_{m+1}) \tag{15}$$

Since the Hessian matrix is symmetric, i.e., $\mathbf{H}^{12}=\mathbf{H}^{21}$, we have:

$$\frac{\partial\mathcal{L}}{\partial\mathbf{k}} = \mathbf{g}_m + \mathbf{g}_{m+1} + (\mathbf{H}^{11}+2\mathbf{H}^{12}+\mathbf{H}^{22})\mathbf{k} - (\mathbf{H}^{11}\mathbf{k}_m + \mathbf{H}^{12}(\mathbf{k}_m+\mathbf{k}_{m+1}) + \mathbf{H}^{22}\mathbf{k}_{m+1}) \tag{16}$$

Setting the derivative to zero yields the optimal condition:

$$(\mathbf{H}^{11}+2\mathbf{H}^{12}+\mathbf{H}^{22})\mathbf{k}^* = \mathbf{H}^{11}\mathbf{k}_m + \mathbf{H}^{12}(\mathbf{k}_m+\mathbf{k}_{m+1}) + \mathbf{H}^{22}\mathbf{k}_{m+1} - (\mathbf{g}_m+\mathbf{g}_{m+1}) \tag{17}$$

**Solving for the Optimal Key Vector $\mathbf{k}^*$**

The optimal key vector $\mathbf{k}^*$ is obtained as:

$$\mathbf{k}^* = (\mathbf{H}^{11}+2\mathbf{H}^{12}+\mathbf{H}^{22})^{-1}(\mathbf{H}^{11}\mathbf{k}_m + \mathbf{H}^{12}(\mathbf{k}_m+\mathbf{k}_{m+1}) + \mathbf{H}^{22}\mathbf{k}_{m+1} - (\mathbf{g}_m+\mathbf{g}_{m+1})) \tag{18}$$

# D  Comparison of AsymKV with Other New Baselines

We also conducted additional comparisons with SnapKV [17], PyramidKV [4], TOVA [20], D2O [26] and L_2-Norm [9].These experiments were performed on the LongBench dataset based on the Llama3-8b-Instruct with compression context max_length=2048.

As show in Table 5 AsymKV still demonstrates performance improvements compared to these baselines.

Table 5: Comparison of AsymKV with other new baselines on LongBench.

| | Single-Doc | Multi-Doc | Sum | Few-shot | Synthetic | Code | Avg. |
|---|---|---|---|---|---|---|---|
| **Llama3-8B-Instruct** | | | | | | | |
| Full Context | 32.19 | 34.59 | 24.96 | 68.48 | 36.96 | 54.41 | 41.46 |
| StreamingLLM | 27.90 | 25.92 | 24.49 | 65.09 | 13.87 | 55.02 | 35.50 |
| LongCache | 28.26 | 25.64 | 24.69 | 65.75 | 15.50 | 54.65 | 35.83 |
| H$_2$O | 30.65 | 32.77 | 24.61 | 61.83 | 37.08 | 54.87 | 39.59 |
| LLMLingua-2 | 26.50 | 30.80 | 24.10 | 39.30 | 22.50 | 32.20 | 29.47 |
| CaM | 30.49 | 31.48 | 24.85 | 63.83 | 37.02 | 55.46 | 39.81 |
| TOVA | 31.82 | 27.94 | 24.57 | 64.34 | 19.29 | 54.06 | 37.04 |
| L2 | 30.18 | 27.41 | 24.70 | 63.29 | 37.34 | 51.78 | 38.43 |
| D2O | 30.81 | 32.87 | 24.64 | 67.42 | 36.67 | 56.49 | 40.85 |
| SnapKV | 32.17 | **34.20** | 25.28 | **68.57** | 37.21 | 53.30 | 41.36 |
| PyramidKV | 31.79 | 34.02 | 25.44 | 68.57 | 37.24 | 54.97 | 41.49 |
| **AsymKV** | **34.45** | 33.64 | **26.17** | 67.94 | **38.66** | **56.61** | **42.32** |

# E   Licenses for Existing Assets

We list the assets used in this paper and their licenses below:

- [24],llama2
- [10],llama3
- [11],Apache 2.0 License
- [29],Apache 2.0 License
- [2],MIT License
- [14],Apache 2.0 License
- [30],CC BY-SA 4.0
- [3],MIT License
- [1],GNU General Public License v3.0
- [6],llama2
- [28],MIT License

