# OpenReview forum: "Homogeneous Keys, Heterogeneous Values: Exploiting Local KV Cache Asymmetry for Long-Context LLMs"
_NeurIPS.cc/2025/Conference — NeurIPS 2025 poster_

### Official Review · Reviewer_AZPM · 2025-06-10

**Clarity:** 3
**Significance:** 2
**Originality:** 2
**Rating:** 3
**Confidence:** 4

**Summary:**

This paper proposes AsymKV, a training-free KV cache compression method for large language models (LLMs) that enables efficient long-context processing by leveraging a previously overlooked asymmetry in attention: keys are locally homogeneous (similar across adjacent positions), while values are heterogeneous. Unlike prior methods that treat keys and values uniformly, AsymKV merges adjacent keys using a second-order optimization technique and preserves values through a mathematically lossless, cardinality-normalized representation. This design minimizes information loss and maintains attention output equivalence. Experiments across various models (e.g., LLaMA3, Mistral) and benchmarks demonstrate that AsymKV provides better accuracy than existing approaches.

**Questions:**

1. How is the proposed method compared with some other KV compression techniques such as TOVA, DuoAttention, and FastGen?
2. What is the compression ratio in Figure 3?
3. Why is the x-axis in Figure 4 denoted as compression ratio? What does it mean when the compression ratio is 100%? Is this a typo?
4. For Table 2, could you explain why almost all KV compression methods seem to have a very good accuracy, compared to the full context baseline, especially on long sequences?

**Ethical Concerns:**

["NO or VERY MINOR ethics concerns only"]

**Final Justification:**

I have read the authors’ rebuttal and increased my score. I will not fight against this paper but the authors should also improve the clarity of some tech details in the final edition. (E.g., clarify the questions raised by reviewers.)

**Limitations:**

yes

**Quality:**

2

**Strengths And Weaknesses:**

[Strengths]
1. The paper empirically validates a key-value asymmetry in attention — keys are homogeneous, values are heterogeneous, and correspondingly proposes a merging method to compress the KV cache.
2. The accuracy of the proposed method is validated across different models and benchmarks, surpassing the existing KV cache compression techniques such as H2O and CaM.


[Weaknesses]
1. Citations and comparisons to some important related works are missing, such as TOVA[1], DuoAttention[2], FastGen[3], and etc.
2. The proposed method, although preserves accuracy better than existing methods such as CaM, does not provide better inference speed than existing solutions.

[1] Matanel Oren, et al. Transformers are multi-state rnns, 2024.

[2] Guangxuan Xiao, et al. Duoattention: Efficient long-context llm inference with retrieval and streaming heads, 2024.

[3] Suyu Ge, et al. Model tells you what to discard: Adaptive KV cache compression for LLMs, 2024.

---

> ### Author Rebuttal · Authors · 2025-07-30
>
> **W1&Q1**: *Comparisons to some important related works are missing, such as TOVA, DuoAttention, FastGen.*
>
> We have added the comparison with SnapKV[1], PyramidKV[2] , **TOVA** [3], D2O[4] and L2-Norm [5]. These experiments are based on the Llama3-8b-Instruct with compression context max_length=2048. AsymKV still demonstrates performance improvements compared to these baselines.
> | LLaMA3-8B-8K       | Single-DocQA | Multi-DocQA | Sum   | Few-shot  Learning | Synthetic | Code   | average |
> |------------|--------------|-------------|-------|-------------------|-----------|--------|---------|
> | TOVA       | 31.82        | 27.94       | 24.57 | 64.34             | 19.29     | 54.06  | 37.04   |
> | L2-Norm         | 30.18        | 27.41       | 24.70 | 63.29             | 37.34     | 51.78  | 38.43   |
> | D2O        | 30.81        | 32.87       | 24.64 | 67.42             | 36.67     | 56.49  | 40.85   |
> | SnapKV \*     | 32.17        | **34.20**       | 25.28 | 68.57             | 37.21     | 53.30  | 41.36   |
> | PyramidKV \*  | 31.79        | 34.02       | 25.44 | **68.57**             | 37.24     | 54.97  | 41.49   |
> | **AsymKV  (ours)**   | **34.45**        | 33.64       | **26.17** | 67.94             | **38.66**     | **56.61**  | **42.32**   |
>
> \* Results are obtained from [2]
>
> [1] SnapKV: LLM Knows What You are Looking for Before Generation
>
> [2] PyramidKV: Dynamic KV Cache Compression based on Pyramidal Information Funneling
>
> [3] Transformers are Multi-State RNNs
>
> [4] D2O: Dynamic Discriminative Operations for Efficient Long-Context Inference of Large Language Models
>
> [5] A Simple and Effective L_2 Norm-Based Strategy for KV Cache Compression
>
> **W2**: *Though preserving accuracy better than methods like CaM, does it fail to outperform existing solutions in inference speed?*
>
> According to Figure 4, AsymKV achieves a comparable accuracy with much fewer tokens than CaM. For example, AsymKV (10% tokens) achieves the effect of CaM (30% tokens). Therefore, if the goal is to achieve higher inference efficiency and lower memory usage while not reducing accuracy, a simple solution is to reduce the token budget in AsymKV.
>
> **Q2**: *What is the compression ratio in Figure 3?*
>
> We would like to clarify that Figure 3 does not involve compression ratio. All methods are uniformly set with context max_length of 2048 tokens. This is explained in lines 206-207.
>
> **Q3**: *Why is the x-axis in Figure 4 denoted as compression ratio? What does it mean when the compression ratio is 100%?*
>
> In line 248, we have already defined the compression ratio as the ratio between the compressed and original token count. Therefore, 100% compression ratio means no compression is applied, and the full context is used.
>
> **Q4**: *For Table 2, could you explain why almost all KV compression methods seem to have a very good accuracy, compared to the full context baseline, especially on long sequences?*
>
> This is a misunderstanding. LongBenchV2 consists of 4-choice questions. Most baselines do not show a significant improvement  over random guesses (25%). In particular, for the Long category (with  lengths ranging from 128k to 2M), all methods except AsymKV have almost degraded to random guesses.

---

> > ### Comment · Reviewer_AZPM · 2025-08-04
> >
> > Thank you for the response. For Q4, could you explain why the performance of AsymKV is prominently better than the Full Context model in Long tasks?

---

> ### Author Response · Authors · 2025-08-04
>
> Thank you for your follow-up question on Q4.
>
> AsymKV serves as a form of regularization in long contexts. Due to the inherent limitations in LLMs' long-context capabilities (e.g., the "lost-in-the-middle" phenomenon [1]), full KV caches often accumulate redundant tokens with low attention scores, thereby diluting the model's focus on relevant information. By selectively merging these less-attended tokens based on attention scores, AsymKV mitigates such contextual noise, leading to more efficient and effective model inference. This regularization effect is also observed in related KV cache optimization works; for instance, Zhang et al. (2024) [2] in H2O observed slight performance gains from evicting low-attention tokens, which they attribute to a similar regularization effect.
>
> [1] Lost in the Middle: How Language Models Use Long Contexts
>
> [2] H2O: Heavy-Hitter Oracle for Efficient Generative Inference of Large Language Models

---

> > ### Author Response · Authors · 2025-08-07
> >
> > Dear Reviewer AZPM,
> >
> > I hope this message finds you well. As the discussion period is drawing to a close with less than two days remaining, I wanted to ensure we have addressed all your concerns satisfactorily, especially your concern about Q4. If there are any additional points or feedback you'd like us to consider, please let us know. Your insights are invaluable to us, and we're eager to address any remaining issues to improve our work.
> >
> > Thank you for your time and effort in reviewing our paper.

---

### Official Review · Reviewer_Tmu7 · 2025-07-02

**Clarity:** 3
**Significance:** 3
**Originality:** 3
**Rating:** 4
**Confidence:** 4

**Summary:**

This paper addresses the phenomenon of asymmetry in KV caches –– adjacent values have distinct heterogeneous distributions, even though adjacent keys are similar in attention weights. Existing methods overlook this asymmetry by merging keys and values identically, while AsymKV instead merges only homogeneous keys using a mathematically derived method that minimizes information loss, and combines the corresponding values via a lossless cardinality-aware normalization to preserve attention outputs exactly. Experiments on LongBench and LongBenchV2 across multiple models (e.g., LLaMA3.1-8B, Mistral-7B) demonstrate that AsymKV outperforms prior state-of-the-art methods like H2O and CaM in both performance and compression efficiency, achieving robust gains even under aggressive compression without retraining.

**Questions:**

1.	In Figure 3, why the result of StreamingLLM is not showing?

2.	According to table 2 and figure 4, why after compression, the results may be better?

**Ethical Concerns:**

["NO or VERY MINOR ethics concerns only"]

**Final Justification:**

The authors have addressed my question about practical on-chip acceleration ratio. I agree with the authors that with a slight change of kernel design, this method could be seamlessly integrated with FlashAttention. I kept my positive score.

**Limitations:**

Yes

**Paper Formatting Concerns:**

No.

**Quality:**

2

**Strengths And Weaknesses:**

## Strengths

1.	The discovery of the heterogeneous distributions of key values is innovative.

2.	The derivation of the best key merging strategy is mathematically rigorous, and ablation study shows its effectiveness.

3.	AsymKV does not require model retraining or fine-tuning, making it practical and easy to adopt across existing LLMs.

4.	AsymKV excels in tasks requiring retention of early context (e.g., Topic Retrieval), where many other methods fail, highlighting its advantage in maintaining semantic continuity.

## Weaknesses

1.	The experiments shows marginal superiority to other compression methods in some tasks, for example “Sum” and “Few-shot”, in table 1. And in some tasks this method is beaten by other baselines.

---

> ### Author Rebuttal · Authors · 2025-07-30
>
> **W2**: *On-chip acceleration may be limited compared to the one with Flash-Attention.*
>
> AsymKV is compatible with the core principle of FlashAttention—leveraging the GPU memory hierarchy (HBM and SRAM), and through tiling, split Q, K, and V into small blocks that are loaded into high-speed SRAM for computation. To adapt the hierarchy to AsymKV, we only need to additionally pass the tiled cardinality vector into SRAM (Eq. (9)).
>
> **Q1**: *In Figure 3, why the result of StreamingLLM is not showing?*
>
> We have shown the results of StreamingLLM in Figure 3. Its line is covered by CAM's line because it is very close to CAM's results.
>
> **Q2**: *According to table 2 and figure 4, why after compression, the results may be better?*
>
> AsymKV acts as a form of regularization: in long contexts, full KV caches often include redundant low-attention tokens that dilute query focus (§3.1). By selectively merging these (based on attention scores), AsymKV reduces noise and quadratic overhead. This phenomenon is also observed in other KV optimization studies; e.g., Zhang et al. (2024) [1] in H2O report slight performance gains from evicting low-attention tokens, attributing it to the regularization effect.
>
> [1] H2O: Heavy-Hitter Oracle for Efficient Generative Inference of Large Language Models

---

> > ### Comment · Reviewer_Tmu7 · 2025-08-05
> >
> > Thank you for your detailed response, all my concerns are resolved. I would like to keep my positive score.

---

> > > ### Author Response · Authors · 2025-08-07
> > >
> > > Thank you for your response and for confirming that all your concerns have been resolved, including our clarifications on **the compatibility with FlashAttention** and **the regularization benefits in KV compression** (explaining why performance can even improve). We truly appreciate your positive evaluation and engagement with our work. We will ensure that these discussed points are incorporated into the final version.

---

### Official Review · Reviewer_B4ak · 2025-07-02

**Clarity:** 3
**Significance:** 3
**Originality:** 3
**Rating:** 5
**Confidence:** 4

**Summary:**

This paper proposes AsymKV, a training-free KV cache compression method that uses some qualitative insights from the authors on the distribution of keys and values in attention mechanisms. The authors observe that the keys of nearby tokens exhibit local homogeneity (i.e., similar attention patterns), while values of nearby tokens can have very different attention distributions. Based on this, they propose two separate KV cache compression strategies: a homogeneity-based merging approach for keys, and a cardinality-aware normalisation for values that preserves attention distributions. AsymKV shows significant improvements compared to baselines like H2O on long-context benchmarks like LongBench.

**Questions:**

How/when do the key-value asymmetries emerge during inference?
How big is the overhead introduced by the backward passes for gradient computations, in terms of time and memory?
A big issue with many KV cache compression methods is the integration with FlashAttention, since it makes materialising attention scores very computationally expensive. Is this a limitation for AsymKV as well?

**Ethical Concerns:**

["NO or VERY MINOR ethics concerns only"]

**Final Justification:**

Thank you for your in-depth answers -- you addressed my main concerns (lack of extremely simple but effective baselines and compatibility with Flash Attention). I'm updating my rating accordingly.

**Limitations:**

Please see my previous comments

**Paper Formatting Concerns:**

Looks very good to me!

**Quality:**

3

**Strengths And Weaknesses:**

The authors did a great job at identifying and characterising the key-value asymmetry in attention distributions; they show on several datasets that attention distributions of the keys of nearby tokens correlate strongly, while this is often not true for values. The framework for deriving the key merging strategy is sound and well-motivated, and LMA also makes sense. The evaluation on several model families (LLaMA, Mistral, Qwen) and benchmarks (LongBenchV2) is comprehensive (although it would be nice to include some straightforward KV cache compression approach like L2 from https://arxiv.org/abs/2406.11430 as a baseline).

I have some concerns about the use of backward passes for gradient computation -- how significant is the overhead it introduces? Does it limit the applicability of the method in memory-constrained settings (which is where KV cache compression makes the most sense)? Can you please expand on this?

Since the whole paper is based on the observation from the authors of key-value asymmetries, it would be nice to have a bit of analysis on those (when/how do they emerge? How do they look like?) -- at the moment this is largely lacking.

---

> ### Author Rebuttal · Authors · 2025-07-30
>
> **W1**: *It would be nice to include some straightforward KV cache compression approach like L2.*
>
> We have added the comparison with SnapKV[1], PyramidKV[2] , TOVA [3], D2O[4] and **L2-Norm** [5]. These experiments are based on the Llama3-8b-Instruct with compression context max_length=2048. AsymKV still demonstrates performance improvements compared to these baselines.
> | LLaMA3-8B-8K       | Single-DocQA | Multi-DocQA | Sum   | Few-shot  Learning | Synthetic | Code   | average |
> |------------|--------------|-------------|-------|-------------------|-----------|--------|---------|
> | TOVA       | 31.82        | 27.94       | 24.57 | 64.34             | 19.29     | 54.06  | 37.04   |
> | L2-Norm         | 30.18        | 27.41       | 24.70 | 63.29             | 37.34     | 51.78  | 38.43   |
> | D2O        | 30.81        | 32.87       | 24.64 | 67.42             | 36.67     | 56.49  | 40.85   |
> | SnapKV \*     | 32.17        | **34.20**       | 25.28 | 68.57             | 37.21     | 53.30  | 41.36   |
> | PyramidKV \*  | 31.79        | 34.02       | 25.44 | **68.57**             | 37.24     | 54.97  | 41.49   |
> | **AsymKV  (ours)**   | **34.45**        | 33.64       | **26.17** | 67.94             | **38.66**     | **56.61**  | **42.32**   |
>
> \* Results are obtained from [2]
>
> [1] SnapKV: LLM Knows What You are Looking for Before Generation
>
> [2] PyramidKV: Dynamic KV Cache Compression based on Pyramidal Information Funneling
>
> [3] Transformers are Multi-State RNNs
>
> [4] D2O: Dynamic Discriminative Operations for Efficient Long-Context Inference of Large Language Models
>
> [5] A Simple and Effective L_2 Norm-Based Strategy for KV Cache Compression
>
> **W2&Q2**: *Concerns about the use of backward passes for gradient  computation -- how significant is the overhead it introduces?*
>
> We have discussed the computational overhead in Section 3.3.2, where we explain the selective nature of our gradient updates, with empirical validation in Table 4.
>
> To elaborate further with a more quantitative perspective, our approach computes gradients **only for the keys in the current chunk**, yielding a gradient tensor of size approximately $c \times d$, where $c$ denotes the chunk size and $d$ is the dimension per token. While backpropagation inherently flows from the output through the model parameters and intermediate states, we mitigate peak memory usage by restricting the `requires_grad` attribute solely to the targeted chunk keys. This ensures that gradients are computed and stored exclusively for those specific tensors, without propagating or materializing gradients for the full model parameters.
>
> By comparison, a standard forward pass stores these parameters plus KV cache states (size $2 \times l \times d$, where $l$ is the maximum sequence length). Thus, our additional overhead is far smaller than the baseline's $O(2 l d + p)$ when $c \ll l$ (typical in chunking), minimizing unnecessary computations across the sequence.
>
> **W3&Q1**: *it would be nice to have a bit of analysis on the observation of key-value asymmetries (when/how do they emerge? How do they look like?)*
>
> While the main paper leverages this observation for compression, we address its characteristics and origins below, with pointers to our existing analyses.
>
> **How Do They Look Like?**
>
> We have provided comprehensive visualizations and quantifications in Appendices A and B, including Spearman correlation heatmaps across multiple layers, heads, models and datasets, revealing strong local key homogeneity (correlations >0.7) versus weak value heterogeneity (correlations <0.4). Notably, in relatively lower layers (1-5), adjacent values tend to exhibit negative correlations.
>
> **How/When Does the Asymmetry Emerge?**
>
> To our knowledge, no prior work explicitly identifies this local KV asymmetry, but its emergence is a natural consequence during transformer pre-training as models balance local attention biases with diverse representations. Key homogeneity arises from strong local dependencies, as shown in Longformer (Beltagy et al., 2020) [1] and Big Bird (Zaheer et al., 2020)[2] , where adjacent keys share high similarity due to syntactic and semantic coherence. Value heterogeneity, in contrast, stems from the need for distinct token information in residual streams, as explored in Synthesizer (Tay et al., 2021) [3] for high-rank attention signals and the Anthropic framework (Elhage et al., 2021) [4] for circuit-level orthogonality. This asymmetry likely solidifies mid-training, optimizing efficiency in attention while preserving expressivity.
>
> [1] Longformer: The Long-Document Transformer, by AI2
>
> [2] Big Bird: Transformers for Longer Sequences, by Google Research
>
> [3] Synthesizer: Rethinking Self-Attention for Transformer Models, by Google Research
>
> [4] A Mathematical Framework for Transformer Circuits, by Anthropic
>
> **Q3**: *Does integration with FlashAttention (expensive for attention scores) limit AsymKV?*
>
> AsymKV is compatible with the core principle of FlashAttention—leveraging the GPU memory hierarchy (HBM and SRAM), and through tiling, split Q, K, and V into small blocks that are loaded into high-speed SRAM for computation. To adapt the hierarchy to AsymKV, we only need to additionally pass the tiled cardinality vector into SRAM (Eq. (9)).

---

> ### Author Response · Authors · 2025-08-07
>
> Dear Reviewer B4ak,
>
> I hope this message finds you well. We are greatly encouraged by your recognition that "_the authors did a great job at identifying and characterising the key-value asymmetry in attention distributions._"
>
> As the discussion period is drawing to a close with less than two days remaining, I wanted to ensure we have addressed all your concerns satisfactorily. If there are any additional points or feedback you'd like us to consider, please let us know. Your insights are invaluable to us, and we're eager to address any remaining issues to improve our work.
>
> Thank you for your time and effort in reviewing our paper.

---

> > ### Author Response · Authors · 2025-08-08
> >
> > Dear Reviewer B4ak,
> >
> > We remain very appreciative of your thoughtful review, especially your encouraging words about our efforts in identifying and characterizing the key-value asymmetry in attention distributions.
> >
> > With the discussion period concluding soon (in under a day), we're circling back to make sure we've fully addressed your feedback. If there's any aspect you'd like us to elaborate on or if you have final thoughts, we'd love to hear them.
> >
> > Thank you once more for your time and expertise.

---

### Official Review · Reviewer_8hqr · 2025-07-02

**Clarity:** 3
**Significance:** 3
**Originality:** 3
**Rating:** 4
**Confidence:** 4

**Summary:**

This paper introduces AsymKV, a novel training-free KV cache compression method tailored for long-context LLM inference. The key insight lies in the empirical discovery of a local asymmetry in attention KV caches: adjacent keys tend to be similar (homogeneous), while adjacent values are highly diverse (heterogeneous). Based on this observation, the authors propose a compression technique that merges adjacent homogeneous keys while preserving the values using cardinality-aware normalization. AsymKV demonstrates consistent improvements over prior methods such as H2O and StreamingLLM across a range of tasks and base models, especially on long-context understanding and early information retrieval.

**Questions:**

1. The paper claims that the method is efficient due to selective gradient computation. However, it would be helpful to include a detailed overhead analysis
2. Can the method be extended to the decoding stage? Given that the outputs of current reasoning models are becoming increasingly long, applying the method only during the prefill stage may be insufficient. While I understand this is beyond the current scope, it would be valuable to hear the authors’ perspective or insights on potential extensions in this direction.

**Ethical Concerns:**

["NO or VERY MINOR ethics concerns only"]

**Final Justification:**

The authors added the results of more baseline methods, as well as several ablation experiments, making the experiments more complementary. Thus I increase my score.

**Limitations:**

Yes

**Quality:**

3

**Strengths And Weaknesses:**

Strength:
1) The discovery of the homogeneous key vs. heterogeneous value asymmetry in KV caches is both original and significant. It challenges prior uniform treatment of keys and values and lays a foundation for more nuanced cache compression strategies in LLMs. It might leads to more exploration of KV Cache compression.
2) The adjacent key merging combined with value addition is a simple yet elegant solution that leverages attention linearity.
3) The authors conduct extensive experiments across multiple base models (LLaMA3.1, Mistral, Qwen2, LLaMA2) and tasks, demonstrating consistent improvements

Weakness:
1) The comparison lacks several state-of-the-art methods such as SnapKV, PyramidKV, and Ada-KV, which have shown strong performance in recent studies. Moreover, in the cache merging category, only CaM is used as a baseline, while other recent methods (e.g., D2O, KVMerger) are omitted, weakening the empirical claims.
2) Some key evaluations, notably compression rate analysis (§4.4) and merge strategy ablations (§4.6), are conducted only on a single dataset or task, which may not be fully representative. This reduces the generalizability of the ablation findings.

[1] https://arxiv.org/abs/2404.14469
[2] https://arxiv.org/abs/2406.02069
[3] https://arxiv.org/abs/2407.11550

---

> ### Author Rebuttal · Authors · 2025-07-30
>
> **W1**: *The comparison lacks several state-of-the-art methods such as SnapKV, PyramidKV, Ada-KV, D2O, and KVMerger.*
>
> We have added the comparison with **SnapKV**[1], **PyramidKV**[2] , TOVA [3], **D2O** [4] and L2-Norm [5]. These experiments are based on the Llama3-8b-Instruct with compression context max_length=2048. AsymKV still demonstrates performance improvements compared to these baselines.
> | LLaMA3-8B-8K       | Single-DocQA | Multi-DocQA | Sum   | Few-shot  Learning | Synthetic | Code   | average |
> |------------|--------------|-------------|-------|-------------------|-----------|--------|---------|
> | TOVA       | 31.82        | 27.94       | 24.57 | 64.34             | 19.29     | 54.06  | 37.04   |
> | L2-Norm         | 30.18        | 27.41       | 24.70 | 63.29             | 37.34     | 51.78  | 38.43   |
> | D2O        | 30.81        | 32.87       | 24.64 | 67.42             | 36.67     | 56.49  | 40.85   |
> | SnapKV \*     | 32.17        | **34.20**       | 25.28 | 68.57             | 37.21     | 53.30  | 41.36   |
> | PyramidKV \*  | 31.79        | 34.02       | 25.44 | **68.57**             | 37.24     | 54.97  | 41.49   |
> | **AsymKV  (ours)**   | **34.45**        | 33.64       | **26.17** | 67.94             | **38.66**     | **56.61**  | **42.32**   |
>
> \* Results are obtained from [2]
>
> [1] SnapKV: LLM Knows What You are Looking for Before Generation
>
> [2] PyramidKV: Dynamic KV Cache Compression based on Pyramidal Information Funneling
>
> [3] Transformers are Multi-State RNNs
>
> [4] D2O: Dynamic Discriminative Operations for Efficient Long-Context Inference of Large Language Models
>
> [5] A Simple and Effective L_2 Norm-Based Strategy for KV Cache Compression
>
>
>
> **W2**: *Some key evaluations (§4.4 and §4.6) are conducted only on a single dataset or task, which may not be fully representative.*
>
> We added the experiment on the summarization task QMSum with the Llama3.1-8B-Instruct. The results are as follows, which are consistent with the conclusions in §4.4 and §4.6.
>
> **§4.4. Compression Rate Analysis.** AsymKV demonstrates superior compression capabilities across all rates.
>
> | Method       | 2%     | 10%    | 20%    | 30%    | 40%    | Full Context |
> |--------------|-------|-------|-------|-------|-------|-------|
> | LongCache    | 18.49 | 19.98 | 20.35 | 20.91 | 21.46 | 23.97 |
> | StreamingLLM | 18.32 | 19.3  | 20.31 | 20.52 | 21.38 | 23.97 |
> | LLMLingua-2   | 17.27  | 19.43  | 20.24  | 21.37  | 20.86  | 23.97  |
> | H2O           | 20.19  | 21.78  | 22.58  | 22.35  | 22.57  | 23.97  |
> | CaM           | 20.12  | 21.69  | 22.39  | 22.42  | 22.62  | 23.97  |
> | AsymKV     | **20.6**  | **22.65** | **23.60** | **24.25** | **24.41** | 23.97 |
>
> **§4.6. Merge Strategy Ablations.** The theoretically-derived optimal key merge strategy consistently  outperforms other approaches. Using distinct strategies for keys and values  is beneficial.
>
> | Method                  | Same as Key | Asymmetric |
> |-------------------------|-------------|------------|
> | Mean Merge              | 19.42       | 21.76      |
> | Weighted by Cardinality | 20.45       | 23.03      |
> | Weighted by Attention   | 18.30       | 22.82      |
> | Optimal k* (Eq. (5))    | **21.04**       | **23.46**      |
>
> **Q1**: *Can detailed overhead analysis of selective gradient computation be added?*
>
> We have discussed the computational overhead in Section 3.3.2, where we explain the selective nature of our gradient updates, with empirical validation in Table 4.
>
> To elaborate further with a more quantitative perspective, our approach computes gradients **only for the keys in the current chunk**, yielding a gradient tensor of size approximately $c \times d$, where $c$ denotes the chunk size and $d$ is the dimension per token. While backpropagation inherently flows from the output through the model parameters and intermediate states, we mitigate peak memory usage by restricting the `requires_grad` attribute solely to the targeted chunk keys. This ensures that gradients are computed and stored exclusively for those specific tensors, without propagating or materializing gradients for the full model parameters.
>
> By comparison, a standard forward pass stores these parameters plus KV cache states (size $2 \times l \times d$, where $l$ is the maximum sequence length). Thus, our additional overhead is far smaller than the baseline's $O(2 l d + p)$ when $c \ll l$ (typical in chunking), minimizing unnecessary computations across the sequence.
>
> **Q2**: *Can the method be extended to the decoding stage?*
>
> We would like to clarify that our method already applies to the decoding stage. See the **Chunk‑wise Parallel Compression** scheme described in Section 3.3.1. After each new chunk of tokens is generated, we trigger the lightweight, parallel compression on the accumulated KV cache—exactly as during prefill—without interrupting token streaming.
> Figure 3 already verified its effect—AsymKV generate long sequences in linear time cost.

---

> ### Author Response · Authors · 2025-08-07
>
> Dear Reviewer 8hqr,
>
> I hope this message finds you well. As the discussion period is drawing to a close with less than two days remaining, I wanted to ensure we have addressed all your concerns satisfactorily. If there are any additional points or feedback you'd like us to consider, please let us know. Your insights are invaluable to us, and we're eager to address any remaining issues to improve our work.
>
> Thank you for your time and effort in reviewing our paper.

---

> ### Comment · Reviewer_8hqr · 2025-08-07
>
> Thanks for your reply!
>
> I think it's more clear to me now, and I will increase my score.

---

### Comment · Area_Chair_9oSD · 2025-08-03
**Reminder: Discussion Phase (July 31 – Aug 6)**

Hi everyone,

This is a reminder that the discussion phase is between July 31 – Aug 6.

Please read the author responses, especially where you are mentioned, and post your reply as soon as possible. This helps ensure there's time for meaningful back-and-forth.

Thanks for your engagement!

AC

---

### Note · Authors · 2025-08-13

We sincerely thank the reviewers for the thoughtful reviews and engaging discussions. We are particularly encouraged by the **unanimous recognition of the originality and significance** of our discovery on homogeneous keys vs. heterogeneous values asymmetry in KV caches, as well as the soundness of our proposed AsymKV method. In particular, Reviewer **B4ak** praised that “*the authors did a great job at identifying and characterising the key-value asymmetry,*” while Reviewer **8hqr** highlighted AsymKV as a “*simple yet elegant solution that leverages attention linearity.*” Reviewer **Tmu7** further commended the “*mathematically rigorous*” derivation.

The reviewers' constructive suggestions primarily focused on empirical comparisons and implementation details, without challenging the core contributions of our discovered phenomenon and method. During the rebuttal phase, we addressed all suggestions comprehensively, including adding comparisons to new baselines (e.g., *SnapKV, PyramidKV, TOVA, D2O, L2-Norm*), providing detailed overhead analysis for selective gradient computation, confirming AsymKV's compatibility with *FlashAttention*, and explaining its regularization effects in long contexts.

We are pleased that these clarifications were positively received: Reviewer **8hqr** explicitly agreed to increase their score, Reviewer **Tmu7** confirmed that all concerns were resolved, and Reviewer **AZPM** engaged in multiple thoughtful rounds of discussion.

We commit to incorporating these enhancements into the revised version of the paper.

Thank you again for your valuable time and insights.

---

### Decision · Program_Chairs · 2025-09-17

**Decision:**

Accept (poster)

**Comment:**

This paper introduces AsymKV, a training-free KV-cache compression method that exploits KV-cache asymmetry for long-context LLM inference. The method is readily integratable with existing solutions such as FlashAttention, offering practical speed-up benefits. The authors have addressed most concerns raised during the rebuttal, and all reviewers are open to accepting the submission. Given the many new results provided in the rebuttal, these should be incorporated into the camera-ready version.